# Pre-Trained Image Encoder for Generalizable Visual Reinforcement Learning

**Zhecheng Yuan**[1], **Zhengrong Xue**[2], **Bo Yuan**[3], **Xueqian Wang**[1],
**Yi Wu**[1,4], **Yang Gao**[1,4], **Huazhe Xu**[1,4]

[1] Tsinghua University [2] Shanghai Jiao Tong University
[3] Qianyuan Institute of Sciences [4] Shanghai Qi Zhi Institute
yuanzc20@mails.tsinghua.edu.cn, xuhuazhe12@gmail.com

## Abstract

Learning generalizable policies that can adapt to unseen environments remains challenging in visual Reinforcement Learning (RL). Existing approaches try to acquire a robust representation via diversifying the appearances of in-domain observations for better generalization. Limited by the specific observations of the environment, these methods ignore the possibility of exploring diverse real-world image datasets. In this paper, we investigate how a visual RL agent would benefit from the off-the-shelf visual representations. Surprisingly, we find that the early layers in an ImageNet pre-trained ResNet model could provide rather generalizable representations for visual RL. Hence, we propose **P**re-trained **I**mage **E**ncoder for **G**eneralizable visual reinforcement learning (PIE-G), a simple yet effective framework that can generalize to the unseen visual scenarios in a zero-shot manner. Extensive experiments are conducted on DMControl Generalization Benchmark, DMControl Manipulation Tasks, Drawer World, and CARLA to verify the effectiveness of PIE-G. Empirical evidence suggests PIE-G improves sample efficiency and significantly outperforms previous state-of-the-art methods in terms of generalization performance. In particular, PIE-G boasts a **55**% generalization performance gain on average in the challenging video background setting. Project Page: https://sites.google.com/view/pie-g/home.

## 1   Introduction

Visual Reinforcement Learning (RL) has achieved significant success in learning complex behaviors directly from image observations [48, 35, 37]. Despite the progress, RL agents are often plagued by the overfitting problem [67], especially in high-dimensional observation space. Previous studies show that it is difficult for the visual agents to generalize to unseen scenarios [8, 40], which severely limits their deployment in real-world applications.

In general, visual RL methods rely on their encoders to learn a visual representation to perceive the world. Recent studies have found that data augmentation [65] leads to more generalizable representations so that the agents can adapt to the unseen environments with different visual appearances [58, 74]. However, most of those approaches only augment the observations of the training environments [37, 39, 68], which is unable to provide enough diversity for generalization over large domain gaps. Furthermore, naively applying data augmentation may damage the robustness of learned representations and decrease training sample efficiency [28, 83].

36th Conference on Neural Information Processing Systems (NeurIPS 2022).

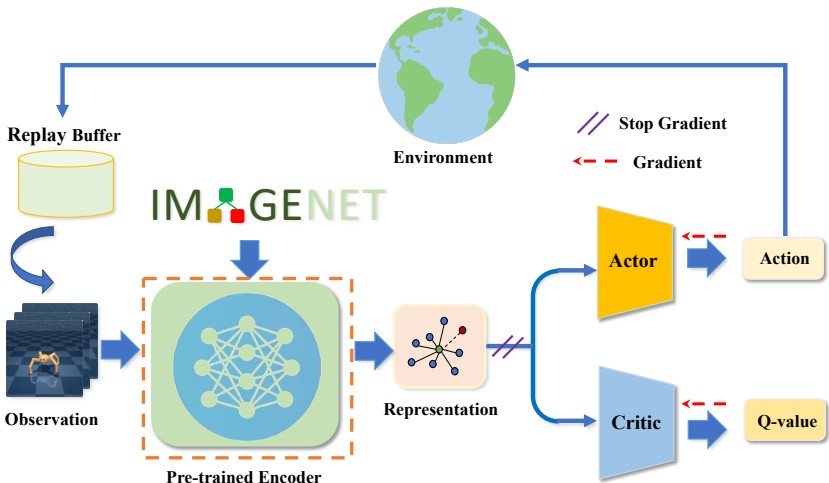

Figure 1: **Overview of PIE-G**. This figure shows the general framework of PIE-G where visual encoders embed high-dimensional images into low-dimensional representations for downstream decision-making tasks. Instead of training the encoder from scratch, PIE-G selects an ImageNet pre-trained ResNet model as the encoder and freezes its parameters during the entire training process.

To overcome these drawbacks, what we require is a universal representation that can generalize to a variety of unseen scenarios. Recent works in representation learning demonstrate promising results in enabling pre-trained models to provide strong priors for downstream tasks [31, 11]. The pre-trained models contain representations obtained from a wide range of existing real-world image datasets. These representations are proved to be robust to noises and capable of distinguishing salient features despite the diversity and the inconsistency [12]. Based on the observations, we would like to ask the following question: is it possible to train a visual RL agent that is augmented with pre-trained visual representations so that it can better generalize to novel tasks?

Towards answering the question, the main contribution of this paper is a surprising discovery that the off-the-shelf features of frozen models trained with ImageNet can be used as universal representations for visual RL. Based on such findings, we present **P**retrained **I**mage **E**ncoder for **G**eneralizable visual reinforcement learning (PIE-G), a visual RL framework that allows agents to obtain enhanced training efficiency and generalization ability via integrating the extracted representations from a pre-trained ResNet [30] encoder into RL training. Straightforward as the framework appears, PIE-G enjoys thoughtful details and nuanced design choices to acquire representations that are suitable for control and generalizable to novel scenarios. Specifically, we show that the choice of early layer features and the ever-updating Batch Normalization (BatchNorm) [33] are crucial for the performance gain.

To validate the effectiveness of our framework, we conduct a series of experiments on 4 benchmarks: DMControl Generalization Benchmark (DMC-GB) [26], DMControl Manipulation Tasks [72], Drawer World [75] that is modified from Meta World [82], and CARLA, a realistic autonomous driving simulator. Empirical studies have demonstrated that PIE-G achieves better or competitive results in training sample efficiency, and significantly outperforms previous state-of-the-art methods in generalization capability without bells and whistles.

Our main contributions are summarized as follows: (i) We find that pre-trained encoders from off-the-shelf image datasets with the early layer features and ever-updating BatchNorm provide generalizable representations in visual RL. (ii) We propose PIE-G, a simple yet effective framework with a pre-trained encoder that can boost the sample efficiency and generalization ability in visual RL. (iii) PIE-G outperforms state-of-the-art methods in 4 visual generalization benchmarks by a large margin, with a **55**% boost on average on the hardest setting in DMC-GB.

## 2   Related Work

**Representation learning in RL.** A large corpus of literature has sought to leverage representation learning in the setting of RL [61, 22, 68, 60, 81, 69]. More recently, there are a branch of methods

that perform unsupervised pre-training to encourage exploration for better sample efficiency [53, 44, 17, 43, 38, 29]. A typical approach is to use a contrastive learning method to jointly incentivize exploration and acquire useful representations [15]. Liu et al. [44] introduce a new type of pre-training techniques via entropy maximization in embedding space for better exploration. In particular, inspired by the new representation algorithm in computer vision, Yarats et al. [79] propose a SwAV-like architecture [4] for pre-training with an exploration scheme that maximizes the entropy of the state visitation distribution. However, all these methods require the data collected in the target environments, resulting in additional sample cost. We instead propose a cross-domain pre-training way to improve visual RL agents' performance without any in-domain interaction during the pre-training time.

**Pre-trained visual encoders for RL.** Applying the pre-trained vision model from other domains to the control tasks has gradually attracted researchers' attention [80, 66, 36, 62, 16, 51, 76]. For example, Shah et al. [63] and Parisi et al. [52] suggest that with the help of experts' demonstrations, the pre-trained ResNet [30] representations can achieve competitive performance with state-based inputs. Moreover, human video datasets are introduced to pre-train a visual representation for downstream policy learning [49]. The pre-trained model has also been proposed for goal-specification via behavior cloning [9]. However, few works explore the effectiveness of pre-trained models for generalization. In contrast to prior approaches, PIE-G enables agents to generalize well to the unseen visual scenarios with a large distributional shift in a zero-shot manner while achieving high sample efficiency in a standard RL training paradigm.

**Generalization in visual RL.** Researchers have investigated to improve visual RL agents' generalization ability from various aspects [1, 27, 40, 73, 2, 50, 77]. Data augmentation [40, 74, 28, 83, 18, 47] and domain randomization [71, 56, 54, 59, 5] are effective ways for generalization in visually different environments. Notably, Hansen et al. [26] employ a BYOL-like[24] architecture to decouple the augmentation from policy learning for better generalization. In order to control the high variance when implementing data augmentation, Hansen et al. [28] add a regularization term of the Q function between un-agumented and augmented data as an implicit variance reduction technique. Meanwhile, Yuan et al. [83] propose a task-aware data augmentation method with the Lipschitz constant [19] for maintaining training stability. Fan et al. [18] apply data augmentation in the imitation learning paradigm. Prior works rely on data augmentation to gain a robust representation for each task. In this work, we tackle this challenge from a different perspective, utilizing a single pre-trained encoder with the universal visual representations for all the tasks.

## 3 Preliminaries

**Reinforcement learning.** Due to partial state observability from images in visual RL [34], we consider learning in a Partially Observable Markov Decision Process (POMDP) [3] formulated by the tuple $\langle \mathcal{S}, \mathcal{O}, \mathcal{A}, r, \mathcal{P}, \gamma \rangle$ where $\mathcal{S}$ is the state space, $\mathcal{O}$ is the observation space, $\mathcal{A}$ is the action space, $r : \mathcal{S} \times \mathcal{A} \mapsto \mathbb{R}$ is a reward function, $\mathcal{P}\left(\mathbf{s}_{t+1} \mid \mathbf{s}_t, \mathbf{a}_t\right)$ is the state transition function, and $\gamma \in [0, 1)$ is the discount factor. The goal is to find a policy $\pi^*$ to maximize the expected cumulative return $\pi^* = \arg\max_\pi \mathbb{E}_{\mathbf{a}_t \sim \pi(\cdot|\mathbf{s}_t), \mathbf{s}_t \sim \mathcal{P}}\left[\sum_{t=1}^{T} \gamma^t r\left(\mathbf{s}_t, \mathbf{a}_t\right)\right]$, starting from an initial state $\mathbf{s}_0 \in \mathcal{S}$ and obtained by following the policy $\pi_\theta\left(\cdot \mid \mathbf{s}_t\right)$ which is parameterized by learnable parameters $\theta$.

**Generalization.** In terms of generalization, we consider a set of POMDPs: $\mathbb{M} = \{\mathcal{M}_1, \mathcal{M}_2, ..., \mathcal{M}_n\}$ that shares the same dynamics and structures. The only difference among them is the observation space $\mathcal{O}$. This setting is more formally described as "Block MDPs" [14]. During the training process, we only have the access to a fixed POMDP denoted $\mathcal{M}_i$. Our purpose is to train an agent on a specific scenario $\mathcal{M}_i$ to learn a policy $\pi_G^*$ which can maximize the expected cumulative return over the whole set of POMDPs in a zero-shot generalization manner.

## 4 Method

In this section, we introduce PIE-G, a simple yet effective framework for visual RL which benefits from the pre-trained encoders on other domains to facilitate sample efficiency and generalization ability.

## 4.1 Pre-trained Encoder

PIE-G explicitly leverages the pre-trained models as the representation extractor without any modification. The pre-trained encoder projects high-dimensional image observations into compact, low-dimensional embeddings that are later used by RL policies. Note that PIE-G is as simple as importing a pre-trained ResNet model from the *torchvision* [46] library. This avoids the design of any auxiliary tasks to acquire useful representations.

For all the training tasks on different benchmarks, the encoder's parameters are frozen to obtain universal visual representations. Since the pre-trained model contains the priors from a wide range of real-world images, we hypothesize that the inherited power from a pre-trained model may help to capture and distinguish the main components of different tasks' observations regardless of the changes of visual appearances or deformed shapes, and will further improve the sample efficiency and generalization abilities of RL agents.

To validate our hypothesis, we first encode each observation independently to obtain embeddings. Then, the embeddings from the second layer of the pre-trained model are fused as input features to the policy networks [64, 51]. Moreover, we enable BatchNorm [33] to keep updating the running mean and running standard deviation during the policy training. The key findings are: 1) early layers of a neural network would provide better representations for visual RL generalization, which resonates with prior works in imitation learning [52]; 2) the always updating statistics in BatchNorm helps better adapt to the shift in observation space and thus improve the generalization ability. More detailed discussion can be found in Section 5.4.

## 4.2 Reinforcement Learning Backbone

We implement DrQ-v2 [78] as the base visual reinforcement learning algorithm. DrQ-v2 is the state-of-the-art method for visual continuous control tasks, which adopts DDPG [41] coupling with clipped Double Q-learning [20] to alleviate the overestimation bias of target Q-value. The agents are trained with two $Q_{\theta_k}$ value functions and their corresponding target network $Q_{\bar{\theta}_k}$. The critic loss function is as follows, and the mini-batch of transitions $\tau = (\mathbf{s}_t, \mathbf{a}_t, r_{t:t+n-1}, \mathbf{s}_{t+n})$ is sampled from the replay buffer $\mathcal{D}$:

$$\mathcal{L}(\theta_k) = \mathbb{E}_{\tau \sim \mathcal{D}} \left[ (Q_{\theta_k}(\mathbf{s}_t, \mathbf{a}_t) - y)^2 \right] \quad \forall k \in \{1, 2\}, \tag{1}$$

with n-step TD target $y$:

$$y = \sum_{i=0}^{n-1} \gamma^i r_{t+i} + \gamma^n \min_{k=1,2} Q_{\bar{\theta}_k}(\mathbf{s}_{t+n}, \mathbf{a}_{t+n}),$$

The actor $\pi_\phi$ is trained with the following objective:

$$\mathcal{L}_\phi(\mathcal{D}) = -\mathbb{E}_{\mathbf{s}_t \sim \mathcal{D}} \left[ \min_{k=1,2} Q_{\theta_k}(\mathbf{s}_t, \mathbf{a}_t) \right], \tag{2}$$

where $\mathbf{s}_t$ is augmented by random shift, $\mathbf{a}_t = \pi_\phi(\mathbf{s}_t) + \epsilon$, $\epsilon$ is sampled from clip $\left( \mathcal{N}\left(0, \sigma^2\right), -c, c \right)$ with a decaying exploration noise $\sigma$.

Thanks to the efficiency of DrQ-v2, PIE-G enjoys faster wall-clock training time and fewer computational footprints. We emphasize that PIE-G does not need any other proprioceptive states and sensory information as the inputs besides the representations extracted from original image observations. In previous works [37, 78, 58, 28], different schemes of data augmentation are proposed to improve sample efficiency and generalization performance. In practice, weak augmentation methods (e.g., random shift) of DrQ [37] and DrQ-v2 [78] are found to be beneficial for sample efficiency. In the setting of generalization, we follow the way of using strong augmentation methods (e.g., mixup) of SVEA [28] and DrAC [58] to further boost the performance. It is worth mentioning that since the gradient is stopped before it reaches the encoder, all the data augmentation techniques discussed here do not affect the pre-trained visual representation. Meanwhile, unlike Rutav et al. [63] and Simone et al. [52], we purely train the agent in a standard RL paradigm without any expert's demonstration.

Table 1: **Generalization on color-jittered observations.** Experiments are conducted on multiple tasks in the DMC-GB (*Top*) and Manipulation Tasks (*Bottom*) environments with varying color backgrounds. For a certain task, the color of the setting in evaluation will be altered. The agent is required to adapt to the changes in a zero-shot manner. Compared with its counterparts, PIE-G gains comparable and better performance in **9** out of **10** settings.

| Setting | DMControl Tasks | SAC | DrQ | DrQ-v2 | SVEA | TLDA | **PIE-G** |
|---|---|---|---|---|---|---|---|
| | Cartpole, Swingup | $248\pm24$ | $586\pm52$ | $277\pm80$ | $\mathbf{837\pm23}$ | $760\pm60$ | $749\pm46$ |
| | Walker, Stand | $365\pm79$ | $770\pm71$ | $413\pm61$ | $942\pm26$ | $947\pm26$ | $\mathbf{960\pm15}$ |
| | Walker, Walk | $144\pm19$ | $520\pm91$ | $168\pm90$ | $760\pm145$ | $823\pm58$ | $\mathbf{884\pm20}$ |
| | Ball_in_cup, Catch | $151\pm36$ | $365\pm210$ | $469\pm99$ | $\mathbf{961\pm7}$ | $932\pm32$ | $\mathbf{964\pm7}$ |
| | Cheetah, Run | $133\pm26$ | $100\pm27$ | $109\pm45$ | $273\pm23$ | $\mathbf{371\pm51}$ | $\mathbf{369\pm53}$ |
| | Manipulation, Training | $2.5\pm1.8$ | $130\pm20$ | $\mathbf{204\pm11}$ | $49\pm48$ | $124\pm32$ | $\mathbf{199\pm13}$ |
| | Modified, Arm | $0.3\pm0.5$ | $68\pm20$ | $29\pm9$ | $21\pm25$ | $55\pm21$ | $\mathbf{122\pm30}$ |
| | Modified, Platform | $0.5\pm0.3$ | $0.8\pm1.3$ | $1.5\pm1.7$ | $24\pm25$ | $89\pm40$ | $\mathbf{96\pm23}$ |
| | Modified, Both | $0.4\pm0.8$ | $1.0\pm2.0$ | $0.8\pm1.5$ | $13\pm14$ | $36\pm25$ | $\mathbf{44\pm16}$ |

# 5 Experiments

In this section, we investigate the following questions: (1) Can PIE-G improve the agent's generalization ability? Specifically, how well does PIE-G deal with jittered color, moving video background, and deformed shapes of robots? (2) Can PIE-G improve training sample efficiency? (3) How do the choice of layers in the encoder and the use of BatchNorm [33] affect the performance? (4) Can further finetuned RL agents outperform those with frozen visual encoders?

## 5.1 Setup

We evaluate our method on a wide range of tasks, including DMControl Generalization Benchmark (DMC-GB) [26], DeepMind Manipulation tasks [72], and Drawer World [75]. PIE-G is trained for 500k interaction steps with 2 action repeat and evaluated with 100 episodes for every task on the testing benchmarks. All the generalization evaluations are in a zero-shot manner. By default, the encoder uses the ResNet18 architecture [30]. To be more specific, the feature maps of the second layer are flattened and passed through an additional fully connected layer to serve as the representations of the observations. More training details and environment descriptions are in Appendix B.

## 5.2 Evaluation on Generalization Ability

We compare the generalization ability of PIE-G with state-of-the-art methods and strong baselines: **SAC** [25]: a widely used off-policy RL algorithm; **DrQ** [37]: a SAC-based visual RL algorithm with augmented observations; **DrQ-v2** [78]: the prior state-of-the-art model-free visual RL algorithm in terms of sample efficiency; **SVEA** [28]: the prior state-of-the-art method in terms of generalization via reducing the Q-variance through an auxiliary loss; **TLDA** [83]: another state-of-the-art method in generalization by using task-aware data augmentation.

**Generalization on color-jittered observations.** The agent's generalization ability is evaluated on DMC-GB with randomly jittered color. For the manipulation tasks, the colors of different objects (e.g., floors, arms) are modified. As shown in Table 1, PIE-G obtains better or competitive performance in **9** out of **10** instances. These results suggest that the visual representation from the pre-trained model is more robust to the color-changing than the one trained by standard RL algorithms.

Table 2: **Generalization on unseen moving backgrounds.** Episode return in two types of unseen dynamic video background environments, i.e., *video easy* (*Bottom*) and *video hard* (*Top*). PIE-G achieves competitive or better performance in **9** out of **12** tasks. In *video hard* setting, we significantly outperforms other algorithms with **+55%** improvement on average.

| Setting | DMControl Tasks | DrQ | DrQ-v2 | SVEA | TLDA | **PIE-G** |
|---|---|---|---|---|---|---|
| | Cartpole, Swingup | 138±9 | 130±3 | 393±45 | 286±47 | **401±21** (+2.0%) |
| | Walker, Stand | 289±49 | 151±13 | 834±46 | 602±51 | **852±56** (+2.2%) |
| | Walker. Walk | 104±22 | 34±11 | 377±93 | 271±55 | **600±28** (+59.2%) |
| | Ball_in_cup, Catch | 92±23 | 97±27 | 403±174 | 257±57 | **786±47** (+95.0%) |
| | Cheetah, Run | 32±13 | 23±5 | 105±37 | 90±27 | **154±17** (+46.6%) |
| | Finger, Spin | 71±45 | 21±4 | 335±58 | 241±29 | **762±59** (+127%) |
| | Cartpole, Swingup | 485±105 | 267±41 | **782±27** | 671±57 | 587±61 |
| | Walker, Stand | 873±83 | 560±48 | 961±8 | **973±6** | 957±12 |
| | Walker. Walk | 682±89 | 175±117 | 819±71 | **873±34** | 871±22 |
| | Ball_in_cup, Catch | 318±157 | 454±60 | 871±106 | 892±68 | **922±20** |
| | Cheetah, Run | 102±30 | 64±22 | 249±20 | **366±57** | 287±20 |
| | Finger, Spin | 533±119 | 456±15 | 808±33 | 744±18 | **837±107** |

**Generalization on unseen and/or moving backgrounds.** We then evaluate PIE-G on the more challenging settings: *video easy* and *video hard* in DMC-GB. The *video hard* setting consists of more complicated and fast-switching video backgrounds that are drastically different from the training environments. Notably, even the reference plane of the ground is removed in this setting.

The comparison results are shown in Table 2. PIE-G achieves better or comparable performance with the prior state-of-the-art methods in **9** out of **12** instances. In particular, PIE-G gains significant improvement in the *video hard* setting over all the previous methods with **+55%** improvement on average. For example, in the *Finger Spin*, *Cup Catch*, and *Walker Walk* tasks, PIE-G outperforms the best of the other methods by substantial margins **127.0%**, **95.0%**, and **59.2%** respectively.

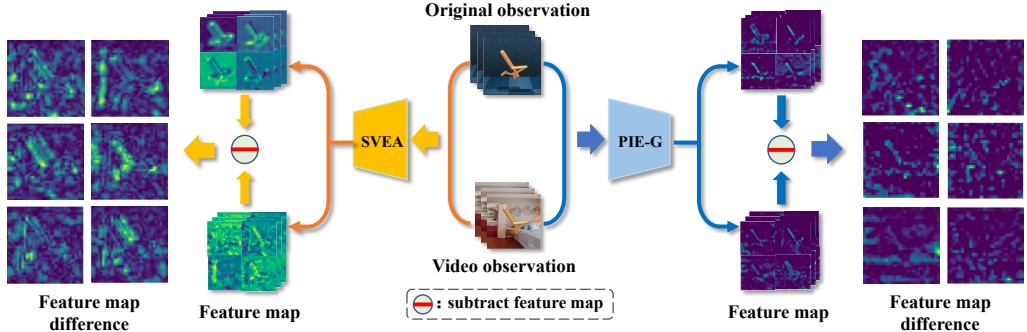

Figure 2: **Visualized feature map differences of two inputs from the same state with different backgrounds.** The difference of the feature maps with PIE-G as the encoder is closer to zero than that with SVEA, indicating PIE-G enjoys better generalization ability.

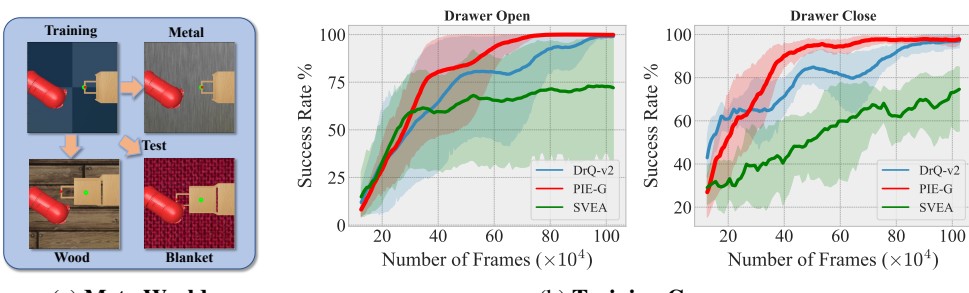

(a) **Meta World**  (b) **Training Curve**

Figure 4: **Training on Meta World.** *Left*: The visualization of Meta World with different textures. *Right*: The training curves. PIE-G *(Red line)* demonstrates better sample efficiency than DrQv2 *(Blue line)* and SVEA *(Green line)*.

Attempting to explain the success, we visualize the difference of the normalized feature maps extracted from the encoder whose inputs are two Walkers of the same pose but with different backgrounds, as is shown in Figure 2. Ideally, a well-generalizable encoder would map the observations of the two Walkers to exactly the same embedding, and therefore the difference should be zero. In practice, as shown in Figure 2, the encoder of PIE-G produces a difference much closer to zero than that of SVEA. Numerically, we calculate the average pixel intensity in the difference of normalized feature maps, and the intensity is decreased by 50.9% with PIE-G than that with SVEA.

Then, we evaluate PIE-G on the CARLA [13] autonomous driving system which contains realistic observations and complex driving scenarios. The default setting is adopted from Zhang et al. [85]. PIE-G and other algorithms are benchmarked on 4 diverse weather environments. As shown in Figure 3, all the prior state-of-the-art methods cannot adapt to the new unseen weather with different lighting, humidity, road conditions etc. The behind reason is that compared to DMC-GB whose images merely consist of a single control agent and the background, the observations of CARLA contain more distracting objects and factors; therefore, only depending on data augmentation to provide diverse data cannot tackle this complicated visual driving task. The experimental results in Figure 3 exhibits that thanks to the ImageNet pre-trained encoder, PIE-G can generalize well on the complicated scenes without large performance drop.

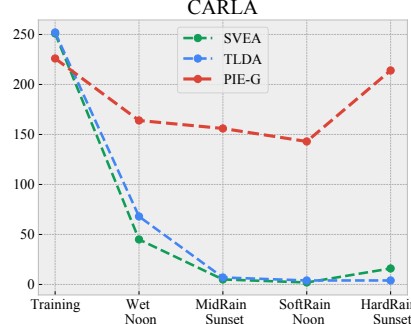

Figure 3: **Performance on CARLA.** PIE-G can well adapt to unseen scenarios.

Furthermore, we conduct experiments on the *Drawer World* benchmark to test the agent's generalization ability in manipulation tasks with different background textures. The visualized observations are shown in Figure 4a. *Success Rate* is adopted as the evaluation metric for its goal-conditioned nature. Table 3 illustrates that PIE-G can achieve better or comparable generalization performance in all the settings with **+24%** boost on average while other approaches may suffer from the CNN's sensitivity in the face of various textures [21].

| Task | Setting | SAC | DrQ-v2 | SVEA | **PIE-G** |
|------|---------|-----|--------|------|-----------|
| Drawer-Close | Training | **100**% | **98**% | 70% | **99**% |
| | Wood | 0% | 32% | 49% | **59**% |
| | Metal | 0% | 46% | 69% | **95**% |
| | Blanket | 0% | 8% | **72**% | 71% |

| Task | Setting | SAC | DrQ-v2 | SVEA | **PIE-G** |
|------|---------|-----|--------|------|-----------|
| Drawer-Open | Training | 98% | **100**% | 75% | 97% |
| | Wood | 18% | 2% | 47% | **79**% |
| | Metal | 35% | 53% | 71% | **97**% |
| | Blanket | 28% | 5% | 37% | **85**% |

Table 3: **Generalization on Drawer World.** Evaluation on distracting textures. PIE-G is robust to the texture changing.

**Generalization on deformed shapes.** To verify agent's robustness in terms of the deformed shapes, we modify the shapes of the jaco arm and the target objects in the manipulation tasks, as shown in Figure 5a. Figure 5b demonstrates that PIE-G also improves the agents' generalization ability with

various shapes while other methods could barely generalize to these changes. We attribute this to the lack of shape changing in previous data augmentation techniques. Conversely, our pre-trained encoder is learned from a multitude of real-world images with various poses and shapes, thus enhancing its generalization ability on deformed shapes.

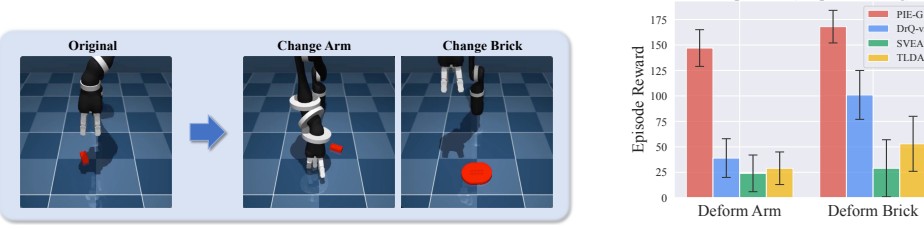

(a) **Deform the shape of the object**     (b) **Generalization performance**

Figure 5: **Deforming the shape.** *Left*: Aiming at evaluating the agent's robustness of the shape, we deform the robot's arm and the target brick. *Right*: The results demonstrate that PIE-G is well-generalizable in the face of deformed shapes.

## 5.3 Evaluation on Sample Efficiency

We evaluate the sample efficiency of PIE-G on 8 relatively challenging tasks on the DeepMind Control Suite and the Manipulation task (*Reach Duplo*). Figure 6 demonstrates that PIE-G achieves better or comparable sample efficiency and asymptotic performance than DrQ-v2 in **7** out of **8** tasks. To eliminate the effect of varied network size, we also include *random enc*, a baseline that has the same network architecture with frozen random initialized parameters. In addition, Figure 4b shows that PIE-G is also superior over the baselines on the Drawer world benchmark. The results demonstrate that the pre-trained encoder inherits the powerful feature extraction ability trained from ImageNet and acquires better training sample efficiency in control tasks.

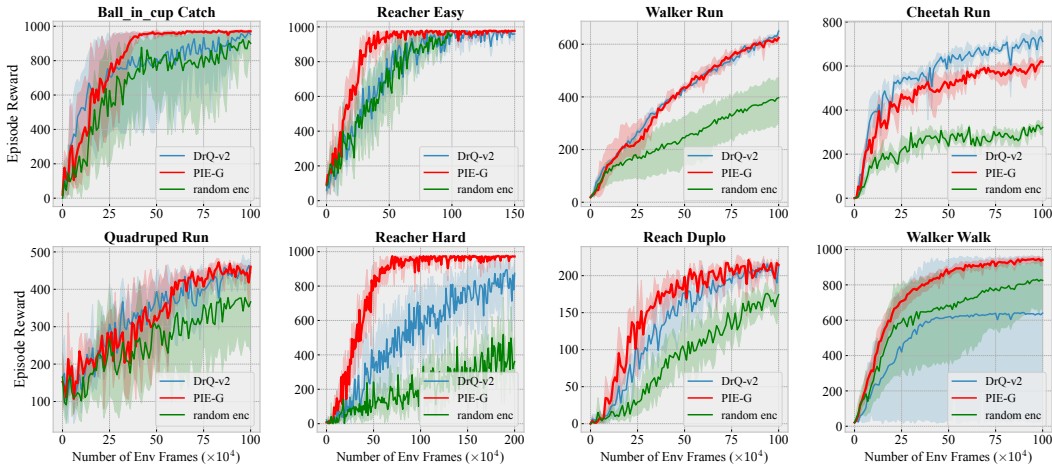

Figure 6: **Training sample efficiency**. Average episode rewards on 8 challenging DMControl tasks with means and standard deviations calculated over 5 seeds. We compare PIE-G (*Red line*) with DrQ-v2 (*Blue line*) and random enc (*Green line*) with respect to sample efficiency. Our method achieves better or comparable performance in **7** out of **8** instances.

## 5.4 Ablation study

To verify the necessity of the design choices in PIE-G, we conduct a series of ablation studies to take a closer look at the proposed method. More results are shown in Appendix C.

**Choice of layers.** In convolutional neural networks, the later layers capture high-level semantic features, while the early layers are responsible for extracting low-level information [45, 84, 42]. Figure 7 and Table 4 investigate how much control tasks can benefit from the features ex-

| Task | Layer 1 | Layer 2 | Layer 3 | Layer 4 |
|------|---------|---------|---------|---------|
| Walker Walk | $840\pm32$ | $\mathbf{884\pm20}$ | $845\pm27$ | $306\pm31$ |
| Cheetah Run | $\mathbf{366\pm56}$ | $369\pm53$ | $294\pm60$ | $111\pm19$ |
| Walker Stand | $953\pm8$ | $\mathbf{964\pm7}$ | $957\pm7$ | $625\pm116$ |

Table 4: **Different layers.** We employ the feature map of different layers of a ResNet model as the visual representation. Among them, the Layer 2 exhibits the best generalization performance.

tracted from different layers. As shown in Figure 7, the early layers preserve rich details of edges and corners, while the later layers only provide very abstract information. Intuitively, for control tasks, a trade-off is required between low-level details and high-level semantics. Table 4 and Figure 8 in Appendix show that the Layer 2 gains better generalization and sample efficiency performance than the other layers.

**Batch normalization.** Batch Normalization (BatchNorm) [33] is a popular technique in computer vision. However, it is not widely adopted in RL algorithms. In contrast to conventional wisdom, BatchNorm is found to be useful and important in PIE-G. Specifically, we find that calculating the mean and variance of the observations during evaluation rather than using the statistics from training data would boost the performance. Figure 9 demonstrates that, in the most challenging settings, PIE-G with the use of BatchNorm can further improve the generalization performance. This is largely because the distribution of observations is determined by the agent, violating the assumption of independent and identical distribution (i.i.d.). This use of BatchNorm also reassures the recommendation

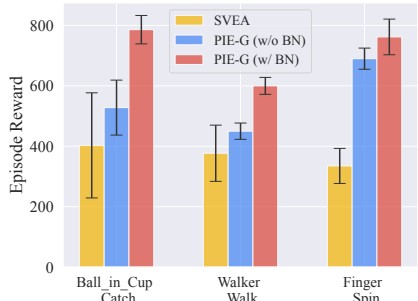

Figure 9: **Leveraging BatchNorm.** The ever-updating BatchNorm is beneficial for better performance.

from Ioffe et al. [33] that recomputation of the statistical means and variances allows the BatchNorm layer to generalize to new data distributions.

**Adopting other datasets for pre-training.** Besides ImageNet [10], we also implement pre-trained visual encoders with other novel and popular datasets: CLIP [57] and Ego4D [23]. CLIP (Contrastive Language–Image Pre-training) trained a large number of (image, text) pairs collected from Internet to jointly acquire visual and text representations. The Ego4D is a

| Tasks | ImageNet | CLIP | Ego4D | SVEA |
|-------|----------|------|-------|------|
| Walker Walk | $600\pm28$ | $615\pm30$ | $441\pm15$ | $377\pm93$ |
| Cheetah Run | $154\pm17$ | $115\pm62$ | $101\pm13$ | $105\pm37$ |
| Walker Stand | $852\pm56$ | $849\pm23$ | $647\pm59$ | $441\pm15$ |
| Finger Spin | $762\pm59$ | $676\pm116$ | $515\pm104$ | $335\pm58$ |

Table 5: **Adopting other datasets for pre-training.** All agents pre-trained with different datasets gain considerable generalization performance.

egocentric human video dataset which contains massive daily-life activity videos in hundreds of scenarios. Table 5 shows that the agents pre-trained with CLIP achieves comparable performance with those pre-trained with ImageNet. Since the Ego4D collects the videos with the *first-person* view, the view difference between the tasks and the dataset leads to a decrease in performance; nevertheless, the Ego4D pre-trained agents still obtain comparable results with the prior state-of-the-art methods.

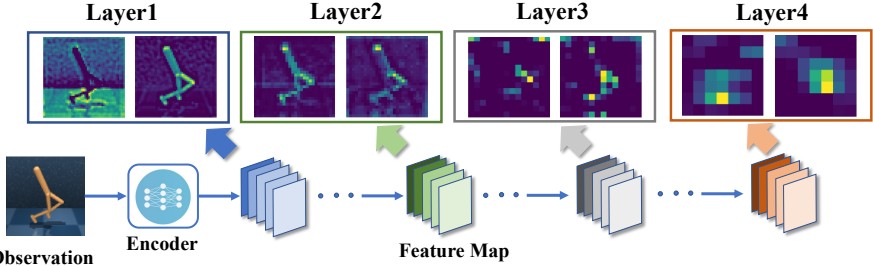

Figure 7: **Visualization of the feature maps of different layers .** The feature map of Layer 2 largely preserves the outline of the Walker that is advantageous to the control tasks, and at the same time discards redundant details.

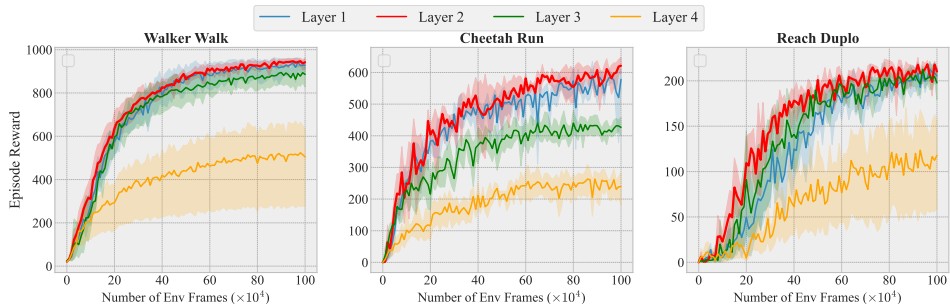

Figure 8: **Choice of layers in terms of sample efficiency.** This figure indicates that the early layers have better sample efficiency than the later layers.

**Finetuning the pre-trained model.** We also conduct research to finetune the encoder's parameters instead of keeping it frozen. Previous works [32, 55] have found that finetuning pre-trained models is challenging. Consistent with these studies, Table 6 suggests that compared with the frozen representations from pre-trained mod-

| Task | PIE-G (Finetune) | PIE-G (Frozen) |
|---|---|---|
| Walker Walk | $455\pm67$ | $\mathbf{600\pm28}$ |
| Cheetah Run | $122\pm15$ | $\mathbf{150\pm19}$ |
| Walker Stand | $771\pm25$ | $\mathbf{852\pm56}$ |

Table 6: **Finetuning the pre-trained models.** We compare the generalization performance between the frozen visual representations and the finetuned ones.

els, the finetuned representations suffer from the out-of-distribution problem [6] and lead to a performance drop in terms of the generalization ability.

**Adopting other pre-trained models.** Additionally, we investigate the efficacy of other visual representations. MoCo-v2 [7] is a pre-trained model optimized via contrastive learning to learn representations. We find that PIE-G with the pre-trained representations of MoCo-v2 can also obtain a comparable performance in terms of

| Task | PIE-G | PIE-G (w / MoCo-v2) |
|---|---|---|
| Walker Walk | $600\pm28$ | $585\pm30$ |
| Cheetah Run | $154\pm17$ | $150\pm19$ |
| Walker Stand | $852\pm56$ | $856\pm51$ |

Table 7: **Adopting other pre-trained models.** PIE-G with MoCo-v2 can obtain comparable generalization performance.

both the sample efficiency and generalization ability. More results are shown in Appendix C.

## 6 Conclusion

In this work, we propose PIE-G, a simple yet effective framework that leverages off-the-shelf features of ImageNet pre-trained ResNet models for better generalization in visual RL. Extensive experiments on a variety of tasks in four RL environments confirm the merits of universal visual representations, which endow the agents with improved sample efficiency and better generalization performance. In addition, we show that the choice of layers and the use of BatchNorm are crucial for the performance gain. Our exploration may inspire more researchers to dig into the great potential of utilizing pre-trained representations in visual RL.

**Limitations.** We study generalization in the simulated environments. However, there might be new challenges in the real-world applications. In the future, we would like to establish and test on benchmarks of real-world scenarios.

**Acknowledgements:** We would like to thank Boyuan Chen, Kaizhe Hu and Guozheng Ma for giving helpful suggestions. We are sincerely grateful to the anonymous reviewers for taking the time to review this work and giving insightful advice. Yang Gao is supported by the Ministry of Science and Technology of the People´s Republic of China, the 2030 Innovation Megaprojects "Program on New Generation Artificial Intelligence" (Grant No. 2021AAA0150000). Yang Gao is also supported by a grant from the Guoqiang Institute, Tsinghua University.

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
