# A Environment Details

**DeepMind control suite.** DMControl Suite [70] is a widely used benchmark, which contains a variety of continuous control tasks. For generalization evaluation, we test methods on the DMControl Generalization Benchmark (DMC-GB) [26] that is developed based on DMControl Suite. DMC-GB provides different levels of difficulty in terms of generalization performance for visual RL. Visualized observations are in the *Setting* column of Table 1 (*Top*) and Table 2.

**DeepMind control manipulation tasks.** DeepMind Control [72] contains dexterous manipulation tasks with a multi-joint Jaco arm and snap-together bricks. In this paper, we modify the colors and shapes of the arms and the bricks in the task of *Reach Duplo* to test the agents' generalization ability. Visualized observations are shown in the *Setting* column of Table 1 (*Bottom*).

**Drawer world benchmarks.** Meta-world [82] contains a series of vision-based robotic manipulation tasks. Wang et al. [75] propose a variant of Meta-world, Drawer World, with a variety of realistic textures to evaluate the generalization ability of the agent. These tasks require a Sawyer robot arm to open or close a drawer, respectively. The visualizations of the environment are shown in Figure 4a.

**CARLA autonomous driving.** CARLA [13] is a realistic simulator for autonomous driving. Many recent works utilize this challenging benchmark in visual RL setting. The trained agents are evaluated on different weather and road conditions.

# B Implementation Details

In this section, we provide PIE-G's detailed settings. As shown in Table 8, we set up our hyper-parmeters and environmental details in three benchmarks. Our method is trained for 500k interaction steps (1000k environment steps with 2 action repeat). All experiments are run with a single GeForce GTX 3090 GPU and AMD EPYC 7H12 64-Core Processor CPU. All code assets used for this project came with MIT licenses. Code: `https://anonymous.4open.science/r/PIE-G-EF75/`

Table 8: Hyperparameter of PIE-G in 4 benchmarks.

| Hyperparameter | DMControl-GB | Drawer World | Manipulation Tasks | CARLA |
|---|---|---|---|---|
| Input size | $84 \times 84$ | $84 \times 84$ | $84 \times 84$ | $84 \times 84$ |
| Discount factor $\gamma$ | 0.99 | 0.99 | 0.99 | 0.99 |
| Action repeat | 8 (cartpole) 2 (otherwise) | 4 | 2 | 2 |
| Frame stack | 3 | 3 | 3 | 3 |
| Learning rate | 1e-4 | 5e-5 | 1e-4 | 1e-4 |
| Random shifting padding | 4 | 4 | 4 | 4 |
| Training step | 500k | 250k | 500k | 500k |
| Evaluation episodes | 100 | 100 | 100 | 50 |
| Optimizer | Adam | Adam | Adam | Adam |

**Vision models.** In this paper, we use the ready-made models from the following link: ResNet: `https://github.com/pytorch/vision`; Moco: `https://github.com/facebookresearch/moco` (v2 version trained with 800 epochs); CLIP: `https://github.com/OpenAI/CLIP`; R3M: `https://github.com/facebookresearch/r3m`.

In terms of implementing data augmentation, We choose *random overlay* (integrate a distract image $\mathcal{I}$ with the observation $o$ linearly, $o' = \alpha o + (1 - \alpha)\mathcal{I}$ as our augmentation method. Similar to the previous works [58, 28] we add a regularization term $\mathcal{R}_\theta$ to the critic objective $\mathcal{F}_\theta$ without introducing extra hyperparameters and other techniques. Our critic loss $\mathcal{J}_\theta$ is as follows, where $\mathcal{D}$ is the replay buffer, $\mathbf{s_t}^{\text{aug}}$ is the augmented observation, $\widehat{Q}(\mathbf{s_t}, \mathbf{a_t}) = r(\mathbf{s_t}, \mathbf{a_t}) + \gamma \mathbb{E}_{\mathbf{s_{t+1}} \sim \mathcal{P}}[V(\mathbf{s_{t+1}})]$:

$$\mathcal{J}_Q(\theta) = \mathcal{F}_Q(\theta) + \mathcal{R}_Q(\theta), \tag{3}$$

with

$$\mathcal{F}_Q(\theta) = \mathbb{E}_{(\mathbf{s_t}, \mathbf{a_t}) \sim \mathcal{D}} \left[ \frac{1}{2} \left( Q_\theta(\mathbf{s_t}, \mathbf{a_t}) - \widehat{Q}(\mathbf{s_t}, \mathbf{a_t}) \right)^2 \right]$$

$$\mathcal{R}_Q(\theta) = \mathbb{E}_{(\mathbf{s_t}, \mathbf{a_t}) \sim \mathcal{D}} \left[ \frac{1}{2} \left( Q_\theta(\mathbf{s_t}^{\text{aug}}, \mathbf{a_t}) - \widehat{Q}(\mathbf{s_t}, \mathbf{a_t}) \right)^2 \right]$$

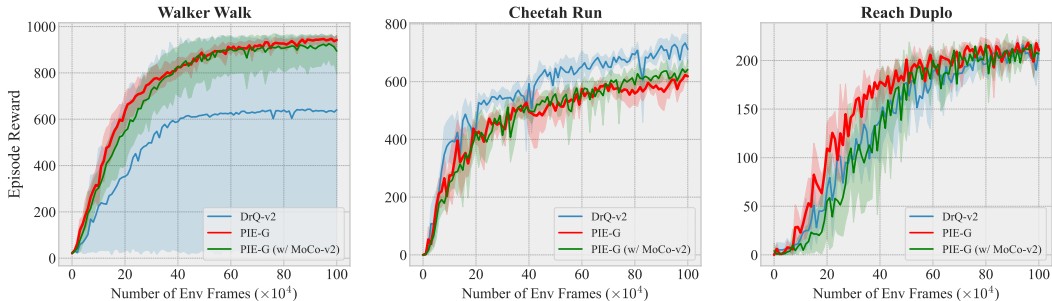

Figure 10: **Other pre-trained models.** PIE-G with MoCo-v2 also achieves competitive sample efficiency.

**Drawer World.** For the Drawer World task, we use a small learning rate in order to maintain the training stability. The episode lengths in Drawer World tasks are 200 steps with 4 action repeat. Random Conv is applied as the data augmentation method. Meanwhile, the original reward setting in the Drawer World is prone to the Q-value divergence. Therefore, we scale the reward by 0.01.

**Manipulation tasks.** The episode lengths in Manipulation tasks are 1000 steps with 2 action repeat. Since for generalization the data augmentation will degrade the training efficiency, we do not apply *random overlay* on this benchmark. We change the physical parameters of *geom size* to deform shape. All methods are evaluated with 100 episodes on different settings.

**CARLA.** We adopt the setting from Zhang et al. [85] (e.g., the reward function and training weather conditions). The maximum episode length in CARLA tasks is 1000 steps with 2 action repeat.

## C    Additional Results

In this section, we provide additional experimental results about PIE-G in various aspects.

### C.1    Comparison with RRL

RRL [63] is another ResNet pre-trained algorithm that can achieve comparable sample efficiency with the state-based algorithms and be robust to the visual distractors. Here we compare the generalization ability of PIE-G with RRL. To compare fairly , we re-implement RRL with DrQ-v2 [78] as the base algorithm which is the state-of-the-art methods in DMControl Suite. Table 9 shows that RRL cannot adapt to the environments with distributional shifts while PIE-G exhibits considerable generalization ability when facing new visual scenarios. We suggest that the choice of layers and ever-updating BatchNorm are the crucial factors for bridging domain gaps and boosting agents' generalization performance.

| Task | RRL | PIE-G |
|---|---|---|
| Walker Walk | $46\pm15$ | $\mathbf{600}\pm\mathbf{28}$ |
| Cheetah Run | $29\pm10$ | $\mathbf{154}\pm\mathbf{17}$ |
| Walker Stand | $154\pm12$ | $\mathbf{856}\pm\mathbf{51}$ |

Table 9: **Compare with RRL.** RRL barely generalize to the new environments in the DMC-GB.

### C.2    Other Pre-trained Models

As shown in Figure 10, PIE-G with the MoCo-v2 pre-trained model also gains a competitive sample efficiency with the help of the off-the-shelf visual representations.

### C.3    Choice of Architectures

We further explore the impact of different network architectures. Since Layer 2 shows better performance, here we choose this layer to extract features.  As shown in Table 10, three kinds of network architectures show comparable generalization performance .  Since ResNet18 is less

| Tasks | ResNet18 | ResNet34 | ResNet50 |
|---|---|---|---|
| Walker Walk | $600\pm28$ | $620\pm38$ | $563\pm57$ |
| Cheetah Run | $154\pm17$ | $143\pm20$ | $149\pm21$ |
| Walker Stand | $852\pm56$ | $867\pm24$ | $871\pm22$ |

Table 10: **Choice of architectures.** PIE-G with different architectures gains comparable generalization performance.

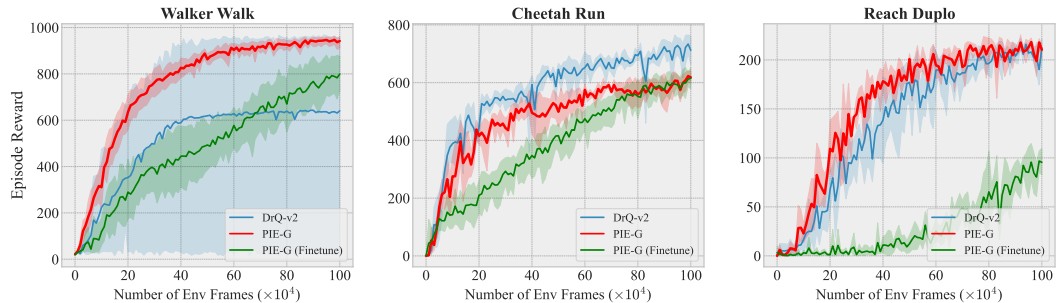

Figure 11: **Finetune the model.** This figure indicates that finetuning the encoder *green line* will sharply reduce the sample efficiency during training.

computationally demanding and with faster wall-clock time than the other two architectures, we choose it as the network backbone.

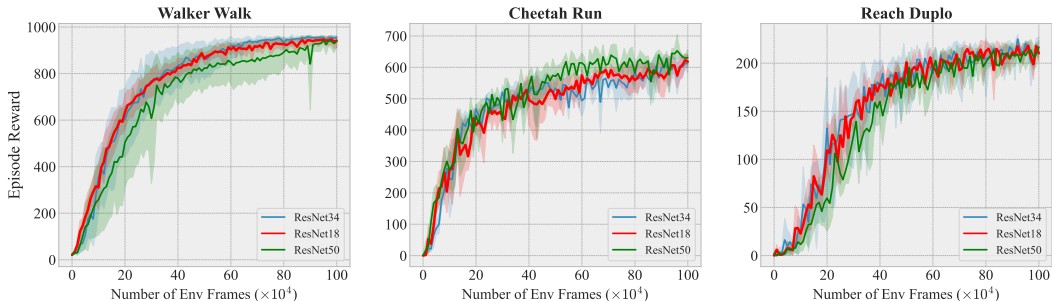

Figure 12: **Choice of architectures.** This figure indicates that PIE-G with various architectures achieves comparable sample efficiency.

## C.4 Finetune Models

As shown in Figure 11, finetuning encoders will significantly reduce sample efficiency. We suggest that during the finetuning process, the encoders have to adapt to the new data distribution and unable to inherit the useful representations learned from the ImageNet, thus severely hindering the improvement in sample efficiency. Additionally, Table 6 and Figure 8 indicate that finetuning the model will make the agents overfit to the training environment.