# OpenReview forum: "Pre-Trained Image Encoder for Generalizable Visual Reinforcement Learning"
_NeurIPS.cc/2022/Conference — NeurIPS 2022 Accept_

### Official Review · Reviewer_ni5r · 2022-07-07

**Rating:** 7
**Confidence:** 4
**Soundness:** 4 excellent
**Presentation:** 4 excellent
**Contribution:** 4 excellent

**Summary:**

The authors investigate using a pre-trained visual backbone in a RL setting. Specifically, they use off-the-shelf ResNet encoders trained in a supervised or self-supervised manner, and use the activations in intermediate layers as input to the policy network, as opposed to the raw pixel values. They find the resulting policy to be significantly more robust to changes in appearance than policies trained on raw pixel values, with fairly dramatic improvements under strongly distracting conditions.

**Questions:**

One question we are left with however, is whether higher-level features (e.g. from levels 3 and 4 of the ResNet) simply underperform because of their lower resolution, or whether the more "semantic" information they encode is actually detrimental for the task. Would it be possible to use these features with higher-resolution images, to control for this difference?

**Limitations:**

No ethical or societal risks here.

**Strengths And Weaknesses:**

The paper presents a simple and effective idea. Although most of the components are standard (pre-trained visual encoders, the policy stacked on top), the authors carefully choose and calibrate these components, and the results are very compelling.

In particular, it is interesting that intermediate feature activations perform better than high-level ones, and that allowing the batch-norm statistics to adapt to the new environment is helpful. These insights make the difference between the method working or not, and will likely apply in a variety of other settings.

---

> ### Author Response · Authors · 2022-08-02
> **Response to reviewer ni5r**
>
> We thank the reviewer for the detailed and thorough review. We added the suggested experiments to the rebuttal revision. In the following, we seek to address your concerns.
>
> ---
>
> **Q:** *"Is whether higher-level features simply underperform because of their lower resolution, or whether the more "semantic" information they encode is actually detrimental for the task. Would it be possible to use these features with higher-resolution images, to control for this difference?"*
>
>
> **A:** Thanks for your suggestions. We conduct additional experiments where we enlarge the resolution of the image observations from 84x84 to 224x224. For the task of *cheetah run*, the generalization performance of the original 84x84 resolution setting with the Layer 2 as the feature is ***369±53*** while the newly conducted 224x224 resolution setting with the Layer 4 as the feature (which has a comparable resolution with the former) is ***213±25***. We conclude that it is not the resolution, but the semantic information that damages the generalization abilities.
>
> It is also worth mentioning that with visual inputs of higher resolution, it will cost more computational resources (2.5x+ CUDA memory) as well as training time (5x+ more than the original one). We added this discussion to the revision of our paper.

---

> ### Author Response · Authors · 2022-08-07
> **Follow-up discussion to Reviewer ni5r**
>
> We would like to first thank you again for your constructive comments and helpful suggestions. Since we are nearly at the end of the discussion phase, we would like to post a follow-up discussion.
>
> In our previous response and revision, we have provided corresponding responses and results, which we believe have covered your concern about the resolution of the input images.
>
> We hope to further discuss with you whether your concerns have been addressed or not. If you still have any unclear parts of our work, please let us know. Thanks.

---

### Official Review · Reviewer_9V7N · 2022-07-08

**Rating:** 3
**Confidence:** 3
**Soundness:** 2 fair
**Presentation:** 3 good
**Contribution:** 1 poor

**Summary:**

Summary: This paper suggests the use of an off the shelf, frozen, pre-trained ResNet model as an encoder for Reinforcement Learning, as opposed to learning an encoder from scratch. The paper suggests that this enables better generalization as the pre-trained ResNet model serves as a universal representation, which is robust and generalizes well to unseen scenarios. They also show that BatchNorm is critical for better generalizing RL agents.

**Questions:**

- Is the fundamental goal to prove that better data distributions are better than better architectures for encoding images in visual RL? That would be a promising direction, but would require significant more experiments than the current manuscript.

- The current experiments show that: (1) Pre-training is better than training from scratch for the same architecture, (2) Simpler architecture with better data performs better than complex architecture with simpler data. Unfortunately, these two are both well known ideas, and building on top of them is necessary to arrive at the point mentioned above.

**Limitations:**

Yes.

**Strengths And Weaknesses:**

Strengths:

1. Generalizing well to unseen environments is a fundamentally important problem, and advances in this direction are both extremely important, and useful for the community.

2. The writing is clear and easy to follow. Figures do a good job of explaining the main idea visually.

Weaknesses:

1. Unfair baseline: The main difference in PIE-G is that the encoder is trained on different data, while the baselines are testing different training methodologies. When evaluating generalization to unseen data, control over the training distribution is absolutely essential. For instance, was the generalization better because the testing environment was maybe overlapping with ImageNet? For the unseen backgrounds case, it appears so as the backgrounds are real-world natural images. The typical game in generalization is to improve without access to new data. Here, new and uncontrolled data is being added to PIE-G, but not to the other baselines. This makes the comparison unfair, and also hard to qualitatively gauge.

2. Unsupported claims: The paper suggests that embeddings from a pre-trained ResNet are a universal representations which are both robust and generalize well. However, it is well documented in the computer vision literature that this is not true [1,2,3]. For instance, works have shown that ImageNet trained models do not generalize to new ImageNet images either [1]. Learning generalizable features is an active area of research, and there is significant evidence suggesting that pretrained ResNet features cannot be called universal, robust or generalizable.

In summary, it appears that the performance gains may be due to the additional dataset which baselines do not have access to, as opposed to learning a universal representation.


References:

1. Recht, B., Roelofs, R., Schmidt, L. and Shankar, V., 2019, May. Do imagenet classifiers generalize to imagenet?. In International Conference on Machine Learning (pp. 5389-5400). PMLR.

2. Hendrycks, D. and Dietterich, T., 2019. Benchmarking neural network robustness to common corruptions and perturbations. arXiv preprint arXiv:1903.12261.

3. He, K., Girshick, R. and Dollár, P., 2019. Rethinking imagenet pre-training. In Proceedings of the IEEE/CVF International Conference on Computer Vision (pp. 4918-4927).

---

> ### Author Response · Authors · 2022-08-02
> **Response to reviewer 9V7N (Question 2-4)**
>
> **Q2:** *"The paper suggests that embeddings from a pre-trained ResNet are universal representations which are both robust and generalize well. However, it is well documented in the computer vision literature that this is not true. "*
>
> **A:** By using “universal”, we mean “multi-purpose with the same encoder”, i.e., we can apply the off-the-shelf ImageNet pretrained encoder with the same parameters to work on multiple RL environments. By contrast, all the other methods train the encoder from scratch for each task and cannot share among tasks. We are by no means saying that it is “almighty/one-fit-for-all”, i.e., it works for all the computer vision tasks without performance drop.
>
> Moreover, we admit that the ImageNet pretrained ResNet model has its intrinsic incompetence, as is stated in your listed references. However, we find out that the representation of this model is competent enough when it serves for learning robust and generalizable policies. Our experiments show that it can concretely improve generalization abilities of the learned policies in a wide range of benchmarks .
>
> ---
>
> **Q3:** *"Is the fundamental goal to prove that better data distributions are better than better architectures for encoding images in visual RL? That would be a promising direction, but would require significantly more experiments than the current manuscript."*
>
> **A:** It is debatable to simply classify ImageNet as the “better data” in visual RL tasks. First, the ImageNet dataset does not contain actions. The way to leverage ImageNet should be carefully designed in decision-making generalization tasks.  Second, there is a domain gap between the  images from ImageNet (real-world images) and the observations from training environments (simulators). Third, as mentioned in **Q1**, in Section 5.4 and Appendix C.1, our experiments demonstrate that with the same training data distribution, only PIE-G can achieve considerable generalization performance. Moreover, existing methods [1,2,5,6] also have shown that naively adding new data will cause training divergence and be detrimental to generalization abilities due to the out-of -distribution problem.
>
> The keypoint is how to leverage new data for visual RL tasks. This is the core of the recent researches. In our paper,  we would like to illustrate that , in contrast to the existing state-of-the-art methods that design auxiliary tasks or more complex training architectures to utilize new data for acquiring representations, we propose a simple yet effective method: levaraging the off-the-shelf representations from ImageNet pre-trained encoder with thoughtful details and nuanced design choices can benefit the agents to significantly improve generalization abilities without bells and whistles.
>
> ---
>
> **Q4:** *"The current experiments show that: (1) Pre-training is better than training from scratch for the same architecture, (2) Simpler architecture with better data performs better than complex architecture with simpler data. Unfortunately, these two are both well known ideas."*
>
> **A:** We want to highlight that the ImageNet pre-training data is out-of-distribution data to the target RL tasks.  Hence, it is unknown and unexplored that whether the data would help and how it can help in visual RL generalization. Many existing studies [12,13,14,15] are still working on it. We are the very first effort to exhibit the power of the ImageNet pre-trained model for generalizing to the unseen visual scenarios.
>
> ---
>
> [12] Khandelwal, Apoorv, et al. "Simple but effective: Clip embeddings for embodied ai." Proceedings of the IEEE/CVF Conference on Computer Vision and Pattern Recognition. 2022.
>
> [13] Simone Parisi, Aravind Rajeswaran, Senthil Purushwalkam, and Abhinav Gupta. The unsurprising effectiveness of pre-trained vision models for control. arXiv preprint arXiv:2203.03580, 2022.
>
> [14] Seo, Younggyo, et al. "Reinforcement learning with action-free pre-training from videos." International Conference on Machine Learning. PMLR, 2022.
>
> [15] Shah, Rutav M., and Vikash Kumar. RRL: Resnet as representation for Reinforcement Learning. International Conference on Machine Learning. PMLR, 2021.

---

> ### Author Response · Authors · 2022-08-02
> **Response to reviewer 9V7N (Question 1)**
>
> Dear reviewer 9V7N, thank you for your detailed and thorough review. In the following, we seek to address each of your concerns.
>
> ---
>
> **Q1:** *"Unfair baseline: when evaluating generalization to unseen data, the typical game in generalization is to improve without access to new data. Here, new and uncontrolled data is being added to PIE-G, but not to the other baselines."*
>
> **A:**  We respectfully disagree with this argument. We explain the reasons in two aspects.
>
> **First, the compared baselines also introduce the new real-world images.** The existing state-of-the-art methods SVEA [1] and TLDA [2] (i.e., the other baselines) are based on RandomOverlay (i.e., Mixup [3]) which overlays the original observations with the images from real-world image datasets (i.e., Places dataset [4]) . A plethora of works [1,2,5,6,7,11] have shown that without adding new data or injecting new visual information, the agents only trained with the observations from the fixed training environment cannot obtain generalization abilities (even generalize to similar tasks [8,9,10]). In our experiments, we have also demonstrated this phenomenon (The column of DrQ, DrQ-v2, SAC in Tables 1,2 of our paper).
>
>    Second, the experiments in Section 5.4 and Appendix C.1 illustrate that compared with **the methods leveraging the same ImageNet pre-trained encoder, only PIE-G can achieve a substantial gain in  generalization performance.**  In Section 5.4, we show that thoughtful designs (frozen parameters, early layers, and ever-updating BatchNorm, etc.) are indispensable to the performance gain; naively importing the pre-trained encoder does not work.  In Appendix C.1, we compare with RRL, a counterpart equipped with the same ImageNet pre-trained ResNet model as the encoder  that can achieve comparable sample efficiency with the state-based algorithms. However,  Table 8 shows that RRL struggles to adapt to new environments.
>
>  In summary, the main goal of visual RL generalization is to find methods that can obtain high performance in unseen environments. The use of additional data is allowed and widely adopted in previous published works [1,2,5,6,7,11]. Moreover, experiments show that naively adding new data would hurt the generalization performance while PIE-G finds a simple and effective way to better leverage these data in comparison to all the baselines.
>
>
>  ---
>
> [1] Nicklas Hansen, Hao Su, and Xiaolong Wang. Stabilizing deep q-learning with convnets and vision    transformers under data augmentation. Advances in Neural Information Processing Systems, 34, 2021.
>
> [2] Zhecheng Yuan et al. Don’t touch what matters: Task-aware lipschitz data augmentation for visual reinforcement learning. arXiv preprint arXiv:2202.09982, 2022.
>
> [3] Zhang, Hongyi, et al. "mixup: Beyond Empirical Risk Minimization." International Conference on Learning Representations. 2018.
>
> [4] B. Zhou, A. Lapedriza, A. Khosla, A. Oliva, and A. Torralba, “Places: A 10 million image database for scene recognition,” IEEE Transactions on Pattern Analysis and Machine Intelligence, 2017. 5, 9
>
> [5] Nicklas Hansen and Xiaolong Wang. Generalization in reinforcement learning by soft data augmentation. In 2021 IEEE International Conference on Robotics and Automation (ICRA), pages 13611–13617. IEEE, 2021.
>
> [6] ​​Linxi Fan, Guanzhi Wang, De-An Huang, Zhiding Yu, Li Fei-Fei, Yuke Zhu, and Animashree Anandkumar. Secant: Self-expert cloning for zero-shot generalization of visual policies. In Proceedings of the 38th International Conference on Machine Learning, PMLR, 2021.
>
> [7]  Kaixin Wang, Bingyi Kang, Jie Shao, and Jiashi Feng. Improving generalization in reinforcement learning with mixture regularization. Advances in Neural Information Processing Systems, 33:7968–7978, 2020.
>
> [8] Song, Xingyou, et al. "Observational Overfitting in Reinforcement Learning." International Conference on Learning Representations. 2019.
>
> [9] Cobbe, Karl, et al. "Leveraging procedural generation to benchmark reinforcement learning." International conference on machine learning. PMLR, 2020.
>
> [10] Farebrother, J., Machado, M. C., and Bowling, M. H. Generalization and regularization in dqn. ArXiv, abs/1810.00123, 2018.
>
> [11] Zhao, Yue, et al. "Intrinsically Motivated Self-supervised Learning in Reinforcement Learning." 2022 International Conference on Robotics and Automation (ICRA). IEEE, 2022.

---

> ### Author Response · Authors · 2022-08-07
> **Follow-up discussion to Reviewer 9V7N**
>
> We would like to first thank you again for your constructive comments and helpful suggestions. Since we are nearly at the end of the discussion phase, we would like to post a follow-up discussion.
>
> In our previous response and our revision, we have provided corresponding responses and results, which we believe have covered your concerns about the fairness of comparing with other baselines, the claim of universal representation,  the keypoint and fundamental goal  of our method,   and our contributions.
>
> We hope to further discuss with you whether your concerns have been addressed or not. If you still have any unclear parts of our work, please let us know. Thanks.

---

> ### Author Response · Authors · 2022-08-09
> **Awaiting your response**
>
> We sincerely thank you for your efforts in reviewing our paper and your suggestions again.
>
> We believe we have resolved all the concerns mentioned in the review. Please let us know if you have further concerns and we are more than happy to address them! Thank you very much !

---

### Official Review · Reviewer_UwhJ · 2022-07-11

**Rating:** 6
**Confidence:** 4
**Soundness:** 4 excellent
**Presentation:** 3 good
**Contribution:** 2 fair

**Summary:**

This paper uses pretrained visual encoders to improve the sample efficiency and generalization performance of visual RL. The main methodological contributions are in the details -- which pretrained layer to transfer, updating of batchnorm parameters, freezing of the encoder, etc. Experiments demonstrate superior performance compared to RL methods that do not use out-of-domain pretraining.

**Questions:**

I think it would be very interesting to delve more into the question of "what kind of pretraining helps for what kind of tasks"? The MoCo comparison at the end of the paper was a step in that direction. More pretrained models could be tried. Are there any lessons to glean? Is it always best just to use the SOTA general-purpose vision system? How then does CLIP do, or its various follow-ups? To me these kinds of experiments would strengthen the originality of the paper, and take it beyond what are, to me, expected results.

Along similar lines, the paper could be strengthened by comparing against a variety of methods that _all use the same pretraining dataset_. It's not terribly surprising that PIE-G can outperform methods that have access to less data than it. A more apples-to-apples comparison would be to compare between methods that all are given access to ImageNet, and then see, among those, which is best, and why. The ablations in Section 5.4 are a step in this direction.

**Limitations:**

I think the discussion of limitations was adequate. There was not much discussion, but I don't think much is needed for this paper.

**Strengths And Weaknesses:**

The main strength of this paper are the results -- it works well! The main weakness, in my opinion, is the originality: many other papers have proposed and demonstrated essentially the same high level point. I will elaborte on these points below:

**Originality**
Making use of powerful pretrained representations is ubiquitous in our field right now (c.f. foundation model hype). This paper characterizes the results as "suprising" but I think the opposite is true: these results will be seen as thoroughly expected by a large portion of the field. This is in part due to the many other papers that have shown similar results in the specific domain of visual pretraining for embodied control (e.g., [76, 72, 48] and many more cited by the paper), and partly because the same trend has been so dominant in other areas of ML. I think [48] does a better job of characterizing this finding, in its title, as "unsurprising". So, the main message of this paper -- that visual pretraining can help, and a lot -- is one that has already, I think, been absorbed by the field (some of the cited related work may be concurrent, but there are also many papers that were published well before the neurips deadline on this topic, including [6], [76], [62], etc). That said, I do think the details in the present paper have some originality, especially the use of batchnorm for adaptation. I personally haven't seen that used before in this specific context (although batchnorm for adaptation in general is a well known method, e.g., https://openreview.net/pdf?id=BJuysoFeg).

**Quality**
The quality of results is impressive: the paper delivers on showing that pretraining helps, both in terms of robustness/generalization and in terms of sample efficiency.

**Clarity**
The writing and figures were all clear to me, except one aspect. I would have liked more detail on how the ResNet was pretrained. This is barely mentioned but seems essential to the success of the method. Indeed I would be interested to see how different pretraining methods and datasets affect the results.

**Significance**
Despite that I don't think readers will be surprised by the results, getting the details right, and achieving high performance with a simple system is a significant accomplishment, and could be impactful assuming the code is released. At the same time, I don't think the specific pretrained model in this paper will be used for long, and subsequent work will have to redo the analysis to pick which layers of future models to transfer, and how to tune the other methodological details.

---

> ### Author Response · Authors · 2022-08-02
> **Response to reviewer UwhJ (Question 4)**
>
> **Q4:** *"Many other papers have proposed and demonstrated essentially the same high level point. Making use of powerful pretrained representations is ubiquitous in our field right now. "*
>
> **A:** We admit that the pre-trained representation has achieved promising results in various domains. However, few works demonstrate that the out-of-domain pre-trained representations could benefit learned agents to generalize to unseen visual scenarios.
>
> In order to further exhibit the effectiveness of PIE-G, we conduct experiments on the CARLA autonomous driving system, which contains realistic observations and complex driving scenarios. Compared to DMC-GB whose images merely consist of a single controllable agent and the background, the observations of CARLA are visually more diverse containing distracting objects, pedestrians, etc.   As shown in the following Table, all the prior state-of-the-art methods cannot adapt to the new unseen weather with different lighting, humidity, road conditions, etc. The experiment results indicate that the prior state-of-the art methods based on data augmentation to diversify data cannot acquire robust representations to tackle such complicated visual driving tasks. In contrast, we propose a new paradigm: utilizing the visual representations from the pre-trained model can improve agent’s generalization abilities on complicated scenes without large performance drop. We added this discussion to the revision of our paper.
>
> | Tasks | PIE-G | SVEA | TLDA |
> | :-----: | :-----: | :-----: | :-----: |
> | Training |  $ 226 \scriptsize \pm 72$   |   $ 251 \scriptsize \pm 22$    |  $ 252 \scriptsize \pm 36$   |
> | WetNoon |  $ 164 \scriptsize \pm 67$    |   $ 45 \scriptsize \pm 44$    |  $ 68 \scriptsize \pm 48$   |
> | SoftRainNoon |  $ 143 \scriptsize \pm 81$    |   $ 2 \scriptsize \pm 2$    |  $ 4 \scriptsize \pm 6$   |
> | MidRainSunset |  $ 156 \scriptsize \pm 97$    |   $ 5 \scriptsize \pm 4$    |  $ 7 \scriptsize \pm 4$   |
>
>
> Furthermore, we provide the theoretical analysis of our generalization problem in visual RL. We quantify that utilizing the encoder with better alignment ability can linearly narrow the generalization gap. We added discussion to the revision of our paper (Appendix C.9).

---

> > ### Comment · Reviewer_UwhJ · 2022-08-07
> > **Thanks for the additional experiments; raising score**
> >
> > Thanks for the thorough response and many additional experiments. The results look pretty great overall. I'm a bit surprised that CLIP did so poorly given it's success in other tasks. It could be interesting to delve into that more -- perhaps CLIP is too invariant geometry. I'd be interested to also see some of the SOTA self-supervised methods added to the camera-ready too (perhaps DINO, etc). I think all the new comparisons really solidify the contribution so I'm raising my score to 6.

---

> ### Author Response · Authors · 2022-08-02
> **Response to reviewer UwhJ (Question 1-3)**
>
> We thank the reviewer for the detailed and thorough review. We added the suggested experiments to the rebuttal revision. In the following, we seek to address each of your concerns.
>
> ---
>
> **Q1:** *"More pretrained models could be tried. Are there any lessons to glean? Is it always best just to use the SOTA general-purpose vision system? How then does CLIP do, or its various follow-ups?"*
>
> **A:** Besides ImageNet, we also implement pre-trained visual encoders with other popular datasets: CLIP and Ego4D. CLIP trained on a large number of (image, text) pairs collected from the Internet to jointly acquire visual and text representations. Ego4D is an egocentric human video dataset which contains massive daily-life activity videos in hundreds of scenarios. The following table shows that the agents pre-trained with CLIP achieve comparable performance with those pre-trained with ImageNet. The ImageNet pre-trained model is empirically slightly better than the CLIP. Since Ego4D collects the videos with the first-person view, the view difference between our target tasks and the Ego4D dataset leads to a decrease in performance; nevertheless, the Ego4D pre-trained agents still obtain comparable results with the prior state-of-the-art methods. We added this discussion to the revision of our paper.
>
>
> | Tasks | ImageNet | CLIP | Ego4D | SVEA |
> | :-----: | :-----: | :-----: | :-----: | :-----: |
> | Walker Walk |  $ 600 \scriptsize \pm 23$   |   $ 615 \scriptsize \pm 30$    |  $ 441 \scriptsize \pm 15$   | $ 377 \scriptsize \pm 93$   |
> | Cheetah Run |  $ 154 \scriptsize \pm 17$    |   $ 115 \scriptsize \pm 62$    |  $ 101 \scriptsize \pm 13$   | $ 105 \scriptsize \pm 37$   |
> | Walker Stand |  $ 852 \scriptsize \pm 56$    |   $ 849 \scriptsize \pm 23$    |  $ 647 \scriptsize \pm 59$   | $ 441 \scriptsize \pm 15$   |
> | Finger Spin |  $ 762 \scriptsize \pm 59$    |   $ 676 \scriptsize \pm 116$    |  $ 515 \scriptsize \pm 104$   | $ 335 \scriptsize \pm 58$   |
>
> ---
>
> **Q2:** *"Along similar lines, the paper could be strengthened by comparing against a variety of methods that all use the same pre-training dataset. A more apples-to-apples comparison would be to compare between methods that all are given access to ImageNet."*
>
> **A:** In Appendix C.1, we compare PIE-G with RRL[1], which is an algorithm leveraging ImageNet pre-trained ResNet as the encoder. RRL can achieve comparable sample efficiency with the state-based algorithms. However, in terms of generalization ability, Table 8 shows that RRL  struggles  to adapt to new visual environments. We believe that this is an apples-to-apples comparison, and it suggests that the designs of PIE-G such as choosing specific layers as well as the ever-updating BatchNorm are the crucial factors for bridging domain gaps and boosting the agents' generalization performance.
>
> [1] Shah, Rutav M., and Vikash Kumar. RRL: Resnet as representation for Reinforcement Learning. International Conference on Machine Learning. PMLR, 2021.
>
> ---
>
> **Q3:** *"I would have liked more detail on how the ResNet was pretrained."*
>
> **A:** In Section 4.1, we mention that PIE-G is as simple as importing a pre-trained ResNet model from the torchvision library The off-the-shelf model provided in torchvision is trained following the method in He et al. [2].   Moreover, all the other pre-trained  models that we use in this paper are unexceptionally off-the-shelf ( links to the source are listed in Appendix.) We did not design any additional auxiliary tasks that may improve the visual encoder’s performance on RL tasks.
>
> [2] Kaiming He, Xiangyu Zhang, Shaoqing Ren, and Jian Sun. Deep residual learning for image recognition. In Proceedings of the IEEE conference on computer vision and pattern recognition, pages 770–778, 2016.

---

> ### Author Response · Authors · 2022-08-07
> **Follow-up discussion to Reviewer UwhJ**
>
> We would like to first thank you again for your constructive comments and helpful suggestions. Since we are nearly at the end of the discussion phase, we would like to post a follow-up discussion.
>
> In our previous response and our revision, we have provided corresponding responses and results, which we believe have covered your concerns about the training with other pre-trained models,  the comparison with another ImageNet pre-trained method, the way of pre-training models, and our contributions.
>
> We hope to further discuss with you whether your concerns have been addressed or not. If you still have any unclear parts of our work, please let us know. Thanks.

---

### Author Response · Authors · 2022-08-02
**General Response**

We thank the reviewers for all the detailed comments and helpful suggestions. We have highlighted the changes in blue in the revised version of our paper. Here we provide an overview of our changes.

(i) Adopting other datasets for pre-training (Appendix C.6). We implement pre-trained visual encoders with other popular CLIP and Ego4D.

(ii) Higher resolution inputs (Appendix C.7). We implement PIE-G with higher resolution images.

(iii) Evaluating on CARLA (Appendix C.8). We evaluate PIE-G on a more challenging and realistic benchmark CARLA.

(iv) Theoretical Analysis (Appendix C.9). we quantify the generalization gap of our zero-shot generalization problem in theory.

---

### Meta-Review · Area_Chair_Tz8J · 2022-08-28

**Recommendation:** Accept
**Confidence:** Certain

**Metareview:**

This paper contains interesting findings in a research topic currently drawing a lot of interest from the community, i.e., RL with pretraining from large-scale general out-of-domain data. I think the use of low-level features and the batch-norm can be interesting to the community. As pointed by reviewer UwhJ, however, I agree that the authors should moderate and clarify some claims in such a way to acknowledge the fact that this line of research has already been studied recently in many works and thus it is not the first finding. I suggest updating the writing in the camera ready version to focus on the specific contributions such as the low-level features and batch-norm. In particular, the contribution (1) in the last paragraph of Introduction is wrong and thus should be changed because it is already well known but not discovered first in this paper.

**Award:**

No

---

### Decision · Program_Chairs · 2022-09-14

Accept